# Psychosocial Outcomes of Individuals Attending a Suicide Bereavement Peer Support Group: A Follow-Up Study

**DOI:** 10.3390/ijerph19074076

**Published:** 2022-03-29

**Authors:** Eve Griffin, Selena O’Connell, Eimear Ruane-McAteer, Paul Corcoran, Ella Arensman

**Affiliations:** 1School of Public Health, University College Cork, T12 XF62 Cork, Ireland; selena.oconnell@ucc.ie (S.O.); eruanemcateer01@qub.ac.uk (E.R.-M.); ella.arensman@ucc.ie (E.A.); 2National Suicide Research Foundation, T12 XF62 Cork, Ireland; pcorcoran@ucc.ie

**Keywords:** suicide, bereavement, peer support, postvention, suicide prevention

## Abstract

Individuals bereaved by suicide represent an important group in terms of postvention. While peer support groups are often accessed by those bereaved, few studies have examined their impact in terms of physical and mental health wellbeing. The aim of this study was to examine psychosocial outcomes of individuals attending suicide bereavement peer support groups in Ireland. Between August 2020 and June 2021, all members were invited to complete a survey, with new members also surveyed at three- and six-month follow-up, to examine changes in wellbeing, depressive symptoms and grief reactions. Results were analyzed using descriptive statistics and mixed linear regression models. The 75 participants were mostly female, with lower levels of overall wellbeing and a higher prevalence of depressive symptoms and suicidal ideation than the general population. Participants also reported high levels of social adjustment difficulties and grief reactions, which were more pronounced for those more recently bereaved. At follow-up (*n* = 28), a significant improvement in wellbeing and a reduction in grief reactions were found, adjusting for time since bereavement. Participants identified the groups as creating a safe space and providing a sense of belonging and hope. Notwithstanding the small number of participants at follow-up, these findings underline the enduring mental health challenges for those bereaved by suicide and provide further evidence for the role of peer support in postvention.

## 1. Introduction

Compared with other types of bereavement, including other types of sudden death, suicide bereavement is in general associated with more negative impacts on emotional, mental and physical health [1,2,3]. Some people may develop long-lasting psychosocial sequalae, including increased risk of suicide and self-harm [2,4]. For every suicide that occurs, 10 people are deeply impacted and up to 135 are affected by the death [5]; therefore, the impacts of suicide can be significant and often wide reaching. Given that grief after suicide tends to be more prolonged and intense, the support and intervention required is likely to be more intensive and specialized than with other types of loss.

Peer support groups specific to suicide bereavement offer compelling appeal to those bereaved by suicide, often over seeking professional assistance. In general, there is growing evidence of the perceived helpfulness of peer support groups for people bereaved by suicide [6,7]. In particular, support groups seem to offer a space where shared experience and understanding among members of the groups provide a unique support [8,9,10]. In many cases, people bereaved by suicide place more confidence in those with similar bereavements, often above medical professionals, feeling that they can understand their particular needs and challenges [9].

However, there are very few studies examining the outcomes of individuals attending suicide bereavement peer support groups [6,11,12,13,14]. To date, only two studies have evaluated such groups. A small study conducted by Barlow and colleagues examined the impact of face-to-face peer support dyads [13]. The findings indicated short-term improvements in aspects of grief reactions among participants measured before and after participation. A more recent study of online support forums found that after 12 months there were significant increases in wellbeing as well as reductions in depressive symptoms [15].

The aim of the current study was to examine the psychosocial wellbeing of individuals attending community-based suicide bereavement peer support groups in Ireland and to determine changes in outcomes at three- and six-month follow-up.

## 2. Materials and Methods

### 2.1. Study Design

A prospective longitudinal study design was used to examine outcomes of individuals attending the peer support groups, using an online survey.

### 2.2. Peer Support Groups

HUGG is a charitable organization that provides community-based peer support groups to adults aged 18 years and older who have been bereaved by suicide (www.hugg.ie (Accessed on 28 March 2022)). The overarching mission of HUGG is to support individuals by reducing isolation and stigma and by promoting resilience and healing. This is achieved by providing evidence-informed peer support groups which promote social connection and coping mechanisms through the lived experience of others.

Each group is led by a trained facilitator and co-facilitator, who have lived experience of suicide bereavement. The meetings have a common format with a structured opening emphasizing the purpose of the meeting and guidelines for participants such as respect, non-judgment and confidentiality to create a safe environment. Following the facilitated group discussion, each meeting is closed in a timely manner with a grounding exercise and a reminder of self-care and other services available. Facilitators may also dedicate some time to sharing information or educational material about grief and bereavement before the meeting is closed. The meeting structure is informed by consultation with guidance from the United Kingdom and Australia [16,17]. At the time this study commenced, there were two operational groups in the wider Dublin area. These groups traditionally met face-to-face on a fortnightly basis, with each session lasting two hours. A maximum number of 12 individuals attended each group. In response to public health restrictions, which were implemented in March 2020 in response to the COVID-19 pandemic, these groups were moved online using video conferencing software. Between August 2020 and June 2021, a further nine groups were established across Ireland.

### 2.3. Participants

All current group members as of August 2020 were invited to participate in a once-off survey. In addition, all new members joining the groups between August 2020 to June 2021 were asked to complete a baseline survey (T1) before attending their first group session. These individuals were also invited to complete two further surveys at three- and six-month follow-up (T2 and T3). All participants were aged over 18 years of age.

### 2.4. Data Items

#### 2.4.1. Demographics

Information on the demographic details recorded included age, gender and marital status. The participants were also asked to provide information on other supports accessed as part of their bereavement, as well as the length of time participating in the support groups. Information on the number of suicide bereavements experienced, the participants’ relationship(s) to the deceased(s) and the month and year of their bereavement(s) were also included in the survey. A series of open-ended questions asked participants about the most and least beneficial aspects of the groups and suggestions for improvements.

#### 2.4.2. Outcome Measures

Overall wellbeing was measured using the World Health Organization-Five Well-being Index (WHO-5) [18,19]. Raw scores range from 0–25 and are multiplied by four to get a total percentage score. It is generally accepted that a score of <50 is indicative of being clinically significant [18].

Depression and suicidal ideation were measured using the Patient Health Questionnaire Depression Scale (PHQ-9) [20]. The PHQ-9 incorporates DSM diagnostic criteria that is used for screening and monitoring depressive symptoms. Scores range from 0–27, with scores of 10 or more suggesting moderate–severe depressive symptoms. Item 9 of the PHQ-9 (‘How often have you been bothered by thoughts that you would be better off dead or of hurting yourself in some way?’) was used to assess suicidal ideation based on the proportion of respondents selecting a response of ‘several days’, ‘more than half the days’ or ‘nearly every day’ within the previous two weeks.

Impaired functioning with regards to social and work settings was measured using the Work and Social Adjustment Scale (SAS-SR) [21]. Scores range from 0–40 with a score of 10 or more indicating impairment of clinical significance.

Traumatic grief was measured using the Traumatic Grief Inventory-Self-Report (TGI-SR) [22,23]. Scores can range from 18–90 and it is thought that scores of 53 or more reflect clinically significant levels of grief.

Two subscales of the Grief Experience Questionnaire (GEQ) were used to measure self-reported somatic reactions and perceived stigma to grief [24,25]. Possible scores range from 5–25 and 11–55, respectively, with higher scores indicating more severe perceptions of stigma and/or somatic reactions. There are no accepted clinical cut-offs for the somatic subscale, while the perceived stigma subscale is generally dichotomized.

### 2.5. Missing Data

Total scores were not computed for the PHQ-9, TGI-SR and GEQ measures where there were three or more items missing or for the SAS-SR and WHO-5 measures where there were two or more missing items. This resulted in data from four participants being excluded at T1. Where fewer items were missing, the mean score of the complete items for that participant was used. There were no data excluded from participants at T2 or T3.

### 2.6. Data Analysis

Descriptive statistics were used to examine the characteristics of participants. Proportions are reported for categorical variables and the mean and standard deviation or median and interquartile range are reported for continuous variables. Differences in categorical variables between groups were examined using chi-square tests. Differences in outcome measures according to time since most recent bereavement were tested using linear regression models, adjusting for age and gender of respondents. The samples were divided into two groups for this analysis: those bereaved within the previous three years and those bereaved for three years or more. Open-ended questions were analyzed using content analysis.

Data from participants who were followed up were analyzed using multilevel linear mixed-effects regression models, estimating the mean change from baseline (T1) to each follow-up (T2 and T3) for all continuous outcome measures. Mixed-effects models use all available data at each time point. A random intercept for individual participants was included in the model to adjust for random heterogeneity in outcome measures between participants. Statistical significance was determined by the *p*-values for the model coefficients. A Bonferroni correction was also applied to provide adjusted *p*-values, which are the original *p*-values multiplied by the number of outcomes examined. Exact McNemar tests were used to explore changes in suicidal ideation (as a categorical variable) from T1 to T2 and from T1 to T3. Analyses were conducted using SPSS Statistics 27.0 (IBM Corp., Armonk, NY) and Stata IC Version 16 (StataCorp, College Station, TX, USA)**.**

## 3. Results

### 3.1. Sample Characteristics

A total of 75 participants completed the baseline surveys; of these, 52 (69.3%) were new group members who completed the baseline survey before attending their first group meeting. The characteristics of these participants are summarized in Table 1. Most were female (*n* = 64; 85.3%) and the median age was 46 years (interquartile range (IQR): 16). Approximately half (*n* = 40; 53.3%) of participants were currently in a relationship or married, with one quarter (19; 25.3%) separated, divorced or widowed.

The median time since bereavement was 1.9 years, ranging from less than 1 month to 51 years. Most of the participants had been bereaved in the previous 3 years (*n* = 45; 60.0%). Approximately one in five had experienced multiple suicide bereavements (*n* = 15; 20.0%). Most participants had lost a spouse or partner to suicide (*n* = 21; 28.0%) or another direct relative (49; 65.3%). All indicated that they had or were currently engaged in other supports for their bereavement, most commonly counseling (*n* = 58; 77.3%), mindfulness therapy (*n* = 15; 20.0%) and writing therapy (12; 16.0%). A small number had attended other peer support groups (*n* = 8; 10.7%). For existing members of the groups (*n* = 23), the median time attending the groups was 9 months and ranged from 1 month to 3 years.

### 3.2. Psychosocial Wellbeing of Respondents

At baseline, most participants recorded negative scores across outcome measures. For wellbeing, more than three quarters of participants reported negative wellbeing scores as measured by the WHO-5 within the clinical range (*n* = 58; 77.3%). A similar proportion (*n* = 59; 78.7%) reported some depressive symptoms as measured by the PHQ-9, with 36 (48.0%) reporting moderate–severe symptoms. Suicidal ideation in the previous two weeks was reported by 18 (24.0%) respondents. More than two thirds (49; 65.3%) of participants reported moderate–severe impairment with regards to social adjustment following their loss and approximately half (*n* = 39; 52.0%) reported clinically significant levels of traumatic grief. The mean score on the somatic subscale of the GEQ was 10.4 (SD: 3.2), while 26.7% (*n* = 20) of participants reported high levels of perceived stigma (see Table 2).

Outcomes varied according to time since most recent bereavement. Compared with those bereaved three or more years, those whose bereavement was within three years had poorer levels of wellbeing (mean diff: −11.8; 95% CI: −21.3 to −2.2; *p* = 0.016), stronger indications of depressive symptoms (+3.5; 0.4 to 6.6; *p* = 0.027), poorer social adjustment (+5.8; 1.3 to 10.4; *p* = 0.013) and higher levels of traumatic grief (+11.6; 4.0 to 19.2; *p* = 0.003). There were no observed differences in terms of somatic grief reactions or perceived stigma (Table 2).

### 3.3. Changes in Outcome Measures at Follow-Up

Of the subset of new members who completed a baseline survey (T1) before attending their first group (*n* = 52), 28 provided at least one follow-up survey (T2 or T3), culminating in 23 completed surveys at both T2 and T3 (44.2%). These participants had attended on average seven sessions after three months (SD: 4.8) and thirteen after six months (SD: 2.8). They did not differ on key demographics (age, time since most recent bereavement, relationship status) or scores on baseline outcome measures. However, males were more likely to participate in follow-up, accounting for 3.4% of new members at T1 but 26.7% of those surveyed at T2 (X^2^ (df) = 5.5 (1); *p* = 0.018).

Adjusting for time since most recent bereavement, a significant improvement in wellbeing was found at T2 (mean difference: +11.8, 95% CI: 4.7 to 18.8), along with a significant reduction in traumatic grief (−6.9, −10.7 to −3.1). All changes held at T3. Changes in somatic reactions to grief, depressive symptoms, social adjustment and perceived stigma were observed, but these did not reach statistical significance (Table 3; Appendix A). No significant changes in the proportion of respondents reporting suicidal ideation at T2 or T3 were observed (Appendix A).

Changes in outcome measures were examined separately for individuals bereaved within the previous three years and individuals bereaved three years or more (Table 4). Improvements in wellbeing held only for those bereaved three years or more (mean change T1 to T2: 13.5, 6.6 to 20.3; *p* < 0.001). In addition, improvements in depressive symptoms between T1 and T2 (−2.5, −5.2 to −0.3) were also observed and some indications of improvements in social adjustment at T2 and T3, for those bereaved three years or more.

### 3.4. Participant Feedback on Group Benefits

Most participants who provided open-ended feedback (*n* = 51) were female (*n* = 40, 78.4%) and had attended the groups for a median of 5 months (IQR: 6). Participants mainly highlighted positive aspects of the groups, which included shared understanding of group members, the group as a safe place to talk, sense of belonging and sense of hope provided by the group, information and advice gained via the group and flexibility to contact group members outside of meetings times (Table 5). A minority of participants suggested challenges or areas for improvements, the most common being to increase the reach/awareness of the groups (*n* = 9, 17.6%).

## 4. Discussion

This is one of the few studies that reports on outcomes of individuals attending a suicide bereavement peer support group. Our findings show that in general those who are bereaved by suicide have poorer mental health than the general population, particularly in relation to general wellbeing, symptoms of depression and suicidal ideation. These mental health impacts were more prevalent for those more recently bereaved. This study makes a unique contribution to the literature by examining the wellbeing of group members up to six months after joining a peer support group. Notable improvements among the relatively small number of participants at three- and six-month follow-ups were observed in wellbeing and indications of traumatic grief.

The demographic profile of participants included in this study is similar to other evaluation studies of peer support groups [13,26]. The majority of participants were immediate family members of the deceased and time since bereavement varied considerably in line with previous research [13,27]. It is difficult to directly compare the baseline scores on the psychosocial indicators with other studies of those bereaved by suicide due to significant variation in the definitions and measures used [28]. However, reported levels of impaired functioning in social and work settings, traumatic grief and perceived stigma were similar to other studies conducted in this area [29,30,31]. The baseline psychosocial measures reported by the respondents of the current study were lower than general population estimates, particularly in relation to overall wellbeing, severity of depressive symptoms and expressions of suicidal ideation [32,33,34]. Psychosocial wellbeing varied according to time since most recent bereavement. Those bereaved more recently, within the past three years, reported poorer wellbeing scores, more severe depressive and somatic symptoms, along with higher levels of traumatic grief and difficulties with social adjustment. All participants reported similar levels of perceived stigma.

While improvements in wellbeing have also been demonstrated in other studies of suicide bereavement peer support [13,15], there are differences in the outcomes showing improvement. For example, a study of users of online peer support forums reported similar improvements on wellbeing at follow-up in addition to reductions in depressive symptoms, but no significant change in traumatic grief [15]. Our study observed changes in wellbeing, traumatic grief and somatic reactions, though there were no significant changes in depressive symptoms, social adjustment or perceived stigma. The current findings suggest that peer support groups may offer different benefits according to how recent the bereavement was, with those more recently bereaved reporting improvements in symptoms of grief rather than overall wellbeing. To our knowledge, how experiences of peer support vary according to time since bereavement have not been examined in the existing literature and future research should consider the mechanisms by which peer support groups work to provide support to individuals according to the time since their bereavement.

Few studies have explored the effective attributes of peer support for suicide bereavement. Drawing on broader peer support literature and theory, perceived benefits include understanding and empathy between peers with similar experiences, social support (emotional and practical) provided by peers, role modeling by peers via experiential knowledge, as well as meaning making through receiving and giving support [7,35,36]. A qualitative study with managers of bereavement peer support programs identified best practices such as ease of accessibility, confidentiality, a safe environment, training and mentoring of peer supporters and close matching of the peer supporters, particularly concerning the bereavement circumstances [36]. The feedback reported in this study is in line with these findings. Broadly positive in their experiences, participants identified the groups as creating a safe space and providing a sense of belonging and hope, as well as providing important peer support even outside the formal group meetings. An ongoing qualitative study builds on these data and explores the experiences of peer support as provided by the HUGG groups and key benefits of this form of support for people bereaved by suicide.

Despite an overall improvement in psychosocial outcomes, the enduring negative wellbeing of participants underlines the need to consider the longer-term impacts of bereavement. In particular, we identified that all participants reported high levels of perceived stigma regardless of time since bereavement, which did not improve at follow-up. This is supported by research which indicates that a suicide bereavement is often associated with increased isolation and social awkwardness in particular [37,38]. While social contact with peers has been identified as one of the most effective interventions for mental-health-related stigma in the short-term, research has not supported its benefits in the long-term [39]. Further research is needed to understand how stigma can be reduced for people bereaved by suicide given that perceived stigma predicts suicidal thoughts and behaviors in this population [31]. Furthermore, the proportion of participants reporting active suicidal ideation did not reduce at follow-up, similar to other research [15], which highlights the ongoing risk of suicidal behavior in those bereaved by suicide. Given that suicide bereavement is associated with a ten-fold risk of suicide [2], services which provide support should consider this ongoing risk and develop appropriate safeguards and signposting to more specialized services as needed. This may involve complementary psychotherapeutic interventions to address the assessment and treatment of suicidal behavior.

### Limitations

There are a number of limitations to this research study. First, we cannot draw firm conclusions on the effects of participation due to the limited sample size at follow-up. Despite the small sample, data completeness at follow-up was high with no missing data on the primary outcome measures. Second, we did not have a control group. To mitigate this limitation, we ensured that baseline surveys were gathered before respondents attended their first group meeting and adjusted for time since most recent bereavement in the analysis. Third, most of the survey respondents were actively engaged with the peer support groups at the time of the research. While the research invitation was circulated to all individuals registered with the peer support groups, we did not have data on rates of disengagement or dropout. Therefore, it is possible that our sample may have more positive attitudes to the peer support groups and indeed may have better coping skills than others who did not engage with the research. Related to this, a fourth limitation is that all participants indicated that they had availed of other supports and treatments, most commonly counseling, which may have contributed to the observed effects. Fifth, the research was undertaken during the COVID-19 pandemic; while usually held in person, the peer support groups moved online in March 2020. The follow-up study involved individuals who had only attended online and any limitations to the online format may have been offset by additional supports such as group text messaging in between meetings.

## 5. Conclusions

It has been flagged in recent years that there are critical gaps in research in the field of suicide bereavement and postvention [40], particularly in relation to evaluation studies. This study has added to the existing evidence supporting the role of suicide bereavement peer support groups, further highlighting the important role of support groups in postvention, while also underlining the enduring mental health challenges related to suicide bereavement.

## Figures and Tables

**Table 1 ijerph-19-04076-t001:** Demographics of survey respondents (*n* = 75).

	*N* (%)
Gender	
Female	64 (85.3)
Male	11 (14.7)
Age in years (Median, IQR)	46 (16)
Current relationship status (missing *n* = 1)	
In a relationship/married	40 (53.3)
Separated/divorced/widowed	19 (25.3)
Single	15 (20.0)
Experienced multiple bereavements to suicide	15 (20.0)
Time since most recent bereavement in years (median, IQR)	1.9 (4.3)
Bereaved less than three years	45 (60.0)
Bereaved more than three years	30 (40.0)
Relationship to deceased	
Spouse/partner	21 (28.0)
Other direct relative	49 (65.3)
Other relationship	5 (6.7)
First time attending support group	
Yes	64 (85.3)
No	11 (14.7)
Other supports accessed	
Counseling	58 (77.3)
Other peer support	8 (10.7)
Bibliotherapy	9 (12.0)
Mindfulness therapy	15 (20.0)
Writing therapy	12 (16.0)

**Table 2 ijerph-19-04076-t002:** Baseline outcome measures and differences according to time since bereavement (*n* = 75).

	All Respondents (*n* = 75; 100%)	Bereaved Less than Three Years (*n* = 45; 60.0%)	Bereaved Three Years or More (*n* = 30; 40.0%)		
	Mean (95% CI)	Mean (95% CI)	Mean (95% CI)	Mean Diff (95% CI)	*p*-Value
Wellbeing (WHO-5)	32.5 (27.3 to 37.7)	27.1 (21.6 to 32.5)	40.4 (30.8 to 50.0)	−11.8 (−21.3 to −2.2)	0.016
Depressive symptoms (PHQ-9)	10.5 (8.9 to 12.1)	11.9 (9.8 to 14.1)	8.3 (5.9 to 10.5)	3.5 (0.4 to 6.6)	0.027
Social adjustment (WSAS)	15.4 (13.1 to 17.8)	17.9 (15.0 to 20.9)	11.7 (8.1 to 15.4)	5.8 (1.3 to 10.4)	0.013
Traumatic grief (TGI-SR)	53.8 (49.8 to 57.7)	58.7 (54.6 to 62.7)	46.7 (39.4 to 53.9)	11.6 (4.0 to 19.2)	0.003
Somatic reactions (GEQ subscale)	10.4 (9.6 to 11.1)	10.3 (9.5 to 11.0)	9.8 (8.5 to 11.1)	0.6 (−0.9 to 2.1)	0.460
Perceived stigma (GEQ subscale)	27.9 (25.7 to 30.2)	27.7 (24.8 to 30.5)	28.5 (24.6 to 32.3)	−1.4 (−5.7 to 2.9)	0.518

**Table 3 ijerph-19-04076-t003:** Change in outcome measures at three-month (T2) and six-month (T3) follow-up.

	T1 M (95% CI)	T2 M (95% CI)	T3 M (95% CI)	Change T1–T2		Change T1–T3	
				Mean Change (95% CI)	*p*-Value (Adjusted)	Mean Change (95% CI)	*p*-Value (Adjusted)
Wellbeing (WHO-5)	36.5 (27.6 to 45.5)	48.3 (38.3 to 58.3)	48.6 (38.6 to 58.6)	+11.8 (4.7 to 18.8)	0.001 (0.006)	+12.1 (4.9 to 19.3)	0.001 (0.006)
Depressive symptoms (PHQ-9)	8.8 (6.1 to 11.5)	7.3 (4.3 to 10.2)	7.3 (4.3 to 10.3)	−1.6 (−3.5 to 0.4)	0.117 (0.702)	−1.5 (−3.5 to 0.5)	0.133 (0.798)
Traumatic grief (TGI-SR)	48.8 (42.4 to 55.2)	41.9 (35.0 to 48.8)	42.3 (35.5 to 49.2)	−6.9 (−10.7 to −3.1)	<0.001 (<0.001)	−6.5 (−10.3 to −2.7)	0.001 (0.006)
Social adjustment (SAS-SR)	12.4 (8.3 to 16.6)	11.1 (6.6 to 15.6)	12.2 (7.7 to 16.7)	−1.4 (−4.0 to 1.3)	0.310 (>1.0)	−0.2 (−2.9 to 2.5)	0.860 (>1.0)
Somatic reactions (GEQ subscale)	9.6 (8.1 to 11.1)	8.3 (6.6 to 10.0)	8.3 (6.6 to 9.9)	−1.3 (−2.5 to −0.1)	0.043 (0.258)	−1.3 (−2.6 to −0.1)	0.037 (0.222)
Perceived stigma (GEQ subscale)	28.0 (23.9 to 32.1)	25.8 (21.3 to 30.3)	25.9 (21.5 to 30.4)	−2.2 (−5.1 to 0.6)	0.127 (0.762)	−2.1 (−5.0 to 0.8)	0.162 (0.972)

**Table 4 ijerph-19-04076-t004:** Change in outcome measures at three-month (T2) and six-month (T3) follow-up according to time since bereavement.

	Bereaved within Previous Three Years				Bereaved Three Years or More			
	Change T1–T2		Change T1–T3		Change T1–T2		Change T1–T3	
	Mean Change (95% CI)	*p*-Value	Mean Change (95% CI)	*p*-Value (Adjusted)	Mean Change (95% CI)	*p*-Value (Adjusted)	Mean Change (95% CI)	*p*-Value (Adjusted)
Wellbeing (WHO-5)	9.2 (−4.2 to 22.6)	0.178 (>1.0)	12.1 (−0.8 to 25.0)	0.066 (0.396)	13.5 (6.6 to 20.3)	<0.001 (<0.001)	11.9 (4.5 to 19.2)	0.002 (0.012)
Depressive symptoms (PHQ-9)	−0.0 (−3.0 to 2.9)	0.988 (>1.0)	−1.1 (−3.9 to 1.9)	0.479 (>1.0)	−2.7 (−5.2 to −0.3)	0.030 (0.180)	−1.9 (−4.6 to 0.7)	0.157 (>1.0)
Traumatic grief (TGI-SR)	−8.3 (−13.7 to −2.9)	0.003 (0.018)	−5.6 (−10.8 to −0.4)	0.035 (0.210)	−5.4 (−10.5 to −0.3)	0.039 (0.234)	−7.1 (−12.6 to −1.7)	0.010 (0.06)
Social adjustment (SAS-SR)	0.9 (−3.8 to 5.8)	0.688 (>1.0)	1.62 (−2.9 to 6.2)	0.491 (>1.0)	−3.3 (−5.9 to −0.6)	0.015 (0.090)	−1.8 (−4.7 to 0.9)	0.198 (>1.0)
Somatic reactions (GEQ subscale)	−2.0 (−3.9 to −0.1)	0.039 (0.234)	−1.0 (−2.9 to 0.9)	0.292 (>1.0)	−0.6 (−2.1 to 0.8)	0.396 (>1.0)	−1.7 (−3.2 to −0.1)	0.038 (0.228)
Perceived stigma (GEQ subscale)	−3.9 (−8.1 to 0.4)	0.076 (0.456)	−1.8 (−5.9 to 2.3)	0.384 (>1.0)	−0.9 (−4.8 to 2.8)	0.607 (>1.0)	−2.5 (−6.5 to 1.6)	0.238 (>1.0)

**Table 5 ijerph-19-04076-t005:** Benefits of peer support groups and frequency of participants reporting.

Benefit	*N* (%)	Description	Supporting Quotation
Shared understanding	35 (68.7)	Participants reported the benefit of speaking to people who had also experienced suicide grief with whom they felt immediately understood and that their experiences were validated.	“The shared experience make the group one of acceptance from the start. The unspoken shared grief through suicide made the group feel like we just all understood.”
Safe place to talk	23 (45.1)	Participants reported the unique space of the groups where they could discuss their experiences without judgement or fear of upsetting others.	“It has given me a chance to talk honestly to people who understand but don’t know me outside of the group, so I’m not worried about their judgment or making them feel bad like I would be if talking to people in my family or friends who might worry about me or feel sad.”
Belonging and connection	16 (31.4)	The sense of belonging to the group and connection to others reduced feelings of isolation.	“I feel part of a group going through the same horrific thing rather than alone.”
Hope and strength	12 (23.5)	Participants felt hope in seeing the progression of others who were further in the grief journey and for some the group was described as a lifeline, particularly during the isolation of the pandemic.	“I would never have been able to cope with the grief alone and it gives me a shimmer of hope that my pain will become manageable.”
Information and advice	11 (21.6)	Participants reported benefit from practical information and resources discussed within the group.	“We all learn from each other and get useful tips and resources from the facilitators.”
Flexibility to contact outside of meeting	11 (21.6)	Participants valued the opportunity to reach out between meetings to facilitators directly or to peers via text messaging group.	“I know at any stage I can reach out for support from the facilitator and members.”

## Data Availability

The data presented in this study are available on request from the corresponding author. The data are not publicly available due to the sensitive nature of the research topic.

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
