# Peer review of "Psychosocial Outcomes of Individuals Attending a Suicide Bereavement Peer Support Group: A Follow-Up Study"

_ijerph, 2022, doi:10.3390/ijerph19074076_

Round 1
Reviewer 1 Report
I would like to thank the authors for their very interesting article. In my opinion, some minor revisions could improve the quality of the article. 1. The summary of results should be more nuanced because the small number of participants at follow-up does not allow to draw firm conclusions on the effects of the participation in the peer support group 2. In the Method section, it would be interesting to propose a more precise description of the meetings: for example, could the authors describe the structured opening and closing? 3. Concerning the characteristics of the sample, the authors should give more details on the relationship with the deceased: how many participants lost a child? a parent? a sibling? 4. The results on the trends in suicidal ideation between T1 and T2/T3 are mentioned in the text but are not reported in Table 3 5. Figure 1 is not very relevant because the results are already presented in Table 3. I think it would be more interesting to replace the Figure with Supplemental Material 1 6. In the Discussion section, the absence of effect on perceived stigma is not discussed, while an effect of the participation on this dimension could have been expected 7. The effects of the participation depending on the time since loss are insufficiently discussed in the Discussion sectionAuthor Response
Please see the attachment

Reviewer 2 Report
I would certainly agree with the authors that peer bereavement programs in suicidology are missing in the literature. This study is an important addition. The study is nicely written and the design and methods are clear. Some interesting results are found.
However, I have a few remarks and suggestions to improve the paper.
- R28-30 This sentence should be mitigated The reference mentioned in (3) also discusses some studies with contra dictionary outcomes. (Use the word In general, (?)
- R44-54. The literature review (2019) of the first author should be mentioned in this paragraph
- Please give an example item
- R118 I am not sure whether you need such an extensive explanation here on missing values. I could have accepted a sentence. Due to item missing values xx persons were not included in the… (Because there are very few). If not then give a reference on the choice of the number of items missing (3) and explain we used the mean score ….of what? (rest of the items on an individual level, the mean score of this particular item and so on?). My suggestion would be not to complicate this too much. It is of minor importance for the results
- R162 If the N=75 then not all the percentages are correct. I could not replicate them for example R165- 166 36 (48,6%). This should be 48%) or the N=74 (probably). I suggest calculating all percentages on N=75.
- R197. In the supplementary material, I suggest a Bonferroni correction. It will not change any outcomes but the procedure is a bit better.
